# Development of a highly specific enzyme-linked immunosorbent assay for detection of antibodies to Duck Tembusu virus using subviral particles

Iyarath Putchong[1], Thaweesak Songserm[2], Sittinee Kulprasertsri[3],
Shintaro Kobayashi[4,5,6], Preeda Lertwatcharasarakul[2], Wallaya Phongphaew[7]*

1 Graduate School of Veterinary Medicine, Faculty of Veterinary Medicine, Kasetsart University, Bangkok, Thailand, 2 Department of Pathology, Faculty of Veterinary Medicine, Kasetsart University, Kamphaeng Saen Campus, Nakhon Pathom, Thailand, 3 Department of Farm Resources and Production Medicine, Faculty of Veterinary Medicine, Kasetsart University, Kamphaeng Saen Campus, Nakhon Pathom, Thailand, 4 Laboratory of Public Health, Faculty of Veterinary Medicine, Hokkaido University, Sapporo, Japan, 5 Veterinary Research Unit, International Institute for Zoonosis Control, Hokkaido University, Kita-ku, Sapporo, Japan, 6 Institute for Vaccine Research and Development (HU-IVReD), Hokkaido University, Kita-ku, Sapporo, Japan, 7 Department of Pathology, Faculty of Veterinary Medicine, Kasetsart University, Bangkok, Thailand

* wallaya.p@ku.th

## Abstract

Duck Tembusu virus (DTMUV) belongs to the family *Flaviviridae* and genus *Orthoflavivirus*. It causes disease in ducks, affecting the nervous system and significantly reducing egg production. The first outbreak of DTMUV in Thailand was reported in 2013, with widespread cases across various regions. However, serological diagnosis of DTMUV is challenging due to antibody cross-reactivity with other flaviviruses. To address this issue, we developed an ELISA based on subviral particles. The cassette encoding the membrane precursor and envelope genes of DTMUV (strain KPS54A61) were cloned into a pCAGGS vector with an OSF-tag and transfected into HEK-293T cells to generate subviral particles. The subviral particles were detected in the supernatant of the transfected cell via immunoblotting using anti-DTMUV E protein and anti-Strep-tag antibodies, which revealed a protein band of approximately 59 kDa. An electron microscopy confirmed the presence of particles approximately 35 nm in diameter. To optimize the SP-based ELISA, checkerboard titration identified the optimal antigen concentration as 70 µg/mL and the optimal serum dilution as 1:100,000. A cut-off value was established for the assay, and testing 300 duck serum samples using the SP-based ELISA identified 41 positive samples (14%) and 259 negative samples (86%). The SP-based ELISA exhibited 100% sensitivity and specificity, achieving a perfect agreement score of 1.0 in comparison with the serum neutralization test. Additionally, specificity testing using antibodies specific to Japanese Encephalitis virus (JEV) revealed no cross-reactivity in the ELISA test. Therefore, the developed SP-based ELISA is highly effective for screening and monitoring DTMUV outbreaks in duck farms, significantly reducing the risk of viral spread and enabling the timely implementation of disease control measures.

**Data availability statement:** All relevant data are within the manuscript and its Supporting information files.

**Funding:** This work was funded by the Office of the Permanent Secretary, Ministry of Higher Education, Science, Research and Innovation (OPS MHESI) under the Research Grant for New Scholar Development Program, grant number RGNS65-038. Additional this work was partially funded by the Faculty of Veterinary Medicine, Kasetsart University, through educational grants and advanced-level graduate research assistantship funding. The funders had no role in study design, data collection and analysis, decision to publish, or preparation of the manuscript.

**Competing interests:** The authors declare that the research was conducted in the absence of any commercial or financial relationships that could be construed as a potential conflict of interest.

## Introduction

Duck Tembusu Virus (DTMUV), a member of the *Flaviviridae* family and *Orthoflavivirus* genus, is a major pathogen affecting waterfowl, particularly ducks. DTMUV infection results in neurological signs, and significant reduction in egg production, leading to substantial economic losses in the duck industry. The first outbreak of DTMUV in Thailand was reported in 2013, with clinical cases identified across various regions, including the northeastern, western, and central areas. During the 2013–2014 period, the mean prevalence of DTMUV infection was recorded at 17.19%, indicating an annual rise in the severity of the outbreak. The morbidity rate ranged between 15–30%, while the mortality rate was observed at 10–15%, depending on farm management practices and the presence of secondary microbial infections. These outbreaks severely impacted Thailand's duck farming industry [1–3]. Additionally, DTMUV can spread to other avian species, including chickens, pigeons, and sparrows [4]. A complete DTMUV particle is approximately 30–60 nanometers in diameter. Similar to other flaviviruses, the genome of DTMUV is positive-sense, single-stranded RNA of approximately 11 kb. This genome contains single open reading frame encoding a polyprotein, which is cleaved into three structural viral proteins, including capsid (C), membrane precursor (prM), and envelope (E) and seven nonstructural (NS) proteins, including NS1, NS2a, NS2b, NS3, NS4a, NS4b, and NS5 [5].

Early detection of DTMUV infection in the flocks is essential for disease control and management. Molecular techniques such as Quantitative Reverse Transcription Polymerase Chain Reaction (RT-qPCR) was employed [6–8]. However, RT-qPCR methods have been considered as an impractical technique for the field epidemiology due to the cost and available equipment [9]. Therefore, serological techniques might be more appropriate for disease screening in field than molecular techniques. Plaque Reduction Neutralization Test (PRNT) [10], and Enzyme-Linked Immunosorbent Assay (ELISA) [11–15], have been widely used in DTMUV diagnosis. PRNT has been considered as a gold standard for detecting antibodies among flaviviruses. However, this technique is time-consuming, laborious technique, and requires live viruses and high-biosafety laboratory level to handle with live viruses. Therefore, ELISA is considered as a rapid and simple technique for detecting antibodies of flaviviruses. However, the serological diagnosis for DTMUV using ELISA was challenged due to the cross-reactivity of the antibodies amongst flaviviruses, including Japanese Encephalitis Virus (JEV) and West Nile Virus (WNV) [16–18].

Recently, an ELISA based on Subviral Particles (SPs) has been developed for highly specific detection of the antibodies against WNV in serum of infected mice. The results showed that the sensitivity of WNV SP-based ELISA was comparable to Serum Neutralization (SN) test with less cross-reactivity to the antibodies against JEV and Tick-borne encephalitis (TBEV) [19,20].

Nonetheless, this technique has not yet been applied to DTMUV; thus, it is possible that the SP-based ELISA method will be employed for the detection of antibodies to DTMUV with high specificity. The present study focuses on the development of a highly specific detection technique for antibody against DTMUV using the SP-based

ELISA. The specificity and sensitivity of the DTMUV-SP-based ELISA will be assessed and compared to the SN, a gold standard technique for the serological detection of flaviviruses.

## Materials and methods

### Sample size calculation

To evaluate the performance of the SP-based ELISA for detecting DTMUV antibodies, the sample size was determined using a prevalence estimation formula for imperfect diagnostic tests, through the Epi-Tool online calculator [21]. The calculation was based on an assumed mean prevalence of 17.19% in Thailand, based on a 2013–2014 study [3], an assumed sensitivity and specificity of 80%, a desired precision of 10%, and a 95% confidence level. The formula used for the calculation, adapted from [22], is:

$$HN = \left(\frac{Z(C)}{L}\right)^2 \times \frac{(HSENS\,(HTP) + [1 - HSPEC]\,[1 - HTP]) \times (1 - HSENS(HTP) - [1 - HSPEC][1 - HTP])}{(HSENS + HSPEC - 1)^2}$$

Where:

- HN (Number of herds sampled) = required sample size,
- Z(C) = Z-score for the desired confidence level (1.96 for 95%),
- L (The tolerance) = desired precision (10% or 0.1),
- HSENS (Herd-level sensitivity) = sensitivity of the test (80% or 0.8),
- HSPEC (Herd-level specificity) = specificity of the test (80% or 0.8),
- HTP (True prevalence of infected herds) = assumed prevalence (17.19% or 0.1719).

The results indicated that at least 225 serum samples would be required. These parameters account for the diagnostic test's imperfections while ensuring reliable prevalence estimation. The calculated sample size provides sufficient data to validate the assay's performance and its utility in diagnosing DTMUV infections in field settings.

### Ethics statement

Serum samples from ducks were collected following the protocol approved by the Institutional Animal Care and Use Committee (IACUC) of Kasetsart University, protocol number ACKU66-VET-009. The studies were conducted in compliance with local legislation and institutional guidelines. Written informed consent was obtained from the owners for the participation of their animals in this study. A total of 300 duck serum samples were collected from duck farms located in Western Thailand. A volume of 5 mL of whole blood was drawn aseptically from the jugular vein of each duck. Blood samples were processed within 4 hours, centrifuged at 2,500 × g for 10 minutes to obtain serum, and stored at −20°C until used. Several serum samples were tested for DTMUV infection using RT-PCR, and these were used as positive (n = 21) and negative (n = 68) controls for the DTMUV-SP-based ELISA. Written consent for the participation of animals was obtained from the owners, ensuring that all procedures were ethically conducted.

### Study design

This study aimed to develop a specific SP-based ELISA to address cross-reactivity issues among flaviviruses, particularly for DTMUV. The ELISA utilized SPs as antigens to enhance specificity and reduce cross-reactivity with other flaviviruses. The assay's performance was evaluated using serum samples from both DTMUV-positive (n = 21) and negative (n = 68)

ducks, previously confirmed by RT-PCR. Statistical methods were employed to determine the cut-off value, and the assay was run in triplicate with optical density (OD) values recorded. A total of 300 serum samples were tested to assess the assay's performance, ensuring its ability to differentiate DTMUV-specific antibodies without interference from other flaviviruses. To assess the specificity of the assay, cross-reactivity with antibodies against other flaviviruses, such as JEV, was tested. Additionally, the sensitivity and specificity of the developed ELISA were validated by comparison with the focus reduction neutralization test (FRNT). A total of 100 serum samples, previously tested by ELISA, were subjected to FRNT in triplicate, and the test agreement between the two methods was calculated to evaluate the reliability and accuracy of the ELISA as a diagnostic tool for detecting antibodies against DTMUV.

## Cells

Human embryonic kidney 293T (HEK-293T; ATCC CRL-3216 ™) and African green monkey kidney (Vero; ATCC CCL-81 ™) cells were grown in high-glucose Dulbecco's modified Eagle's medium with 20 mM L-glutamine (DMEM; Invitrogen, MA, USA) supplemented with 10% fetal bovine serum (FBS; Invitrogen, MA, USA), and 1% Antibiotic-Antimycotic (Thermo Fisher Scientific, MA, USA), and growth at 37°C with supplied 5% $CO_2$.

## Construction of the plasmid encoding the cassette of recombinant DTMUV_prM-OSF-E

The DNA fragment encoding the DTMUV prM-E protein was synthesized based on the optimized cDNA sequence of the Thai field strain DTMUV (strain KPS54A61), with recognition sequences for *XhoI* (NEB, MA, USA) and *EcoRI* (NEB, MA, USA) added at the 5′ and 3′ ends, respectively. Additionally, One-STrEP-Flag (OSF) tags were inserted between the prM and E genes using the Gibson Assembly® Kit (NEB, MA, USA). The DTMUV_prM-OSF-E cassette was subsequently inserted into a pCAGGS plasmid via the *XhoI* and *EcoRI* restriction sites. The resulting recombinant plasmid was designated pCAGGS_DTMUV_prM-OSF-E.

## Production of DTMUV-subviral particles (SPs) and harvest

The pCAGGS_DTMUV_prM-OSF-E was transfected into HEK-293T cells using Polyethylenimine "Max" (PEI MAX®; MW 40,000; Polysciences, PA, USA), transfecting reagent, following the manufacturer's instruction. The transfected cells were incubated for 72 hrs. at 37°C, supplemented with 5% $CO_2$. The cells and supernatant were collected at 48 and 72 hrs. post-transfection. The transfected cells were lysed using a lysis buffer (50 mM Tris HCl pH 7.5, 1% Triton X-100, 1 mM EDTA, and 0.25 M sucrose). Meanwhile, the supernatant was subjected to polyethylene glycol 8000 (PEG 8000; Glentham, Corsham, UK) precipitation. For PEG precipitation, the supernatant was mixed with PEG 8000 and NaCl in the final concentrations 10% and 1.9%, respectively, and incubated for 16 hrs. at 4°C. Subsequently, the SPs were harvested by centrifugation at 11,000 × g for 20 mins. and then resuspended with phosphate buffered saline (PBS) and stored at −80°C until used.

## Characterization of DTMUV-SPs

**SDS-PAGE and western blotting.** The supernatant and lysed cells were analyzed for the presence of the SPs using sodium dodecyl sulfate polyacrylamide gel electrophoresis (SDS-PAGE) and western blotting techniques. The SPs were subjected into 10% polyacrylamide gel, and then transferred to a nitrocellulose membrane, washed with PBS-T (0.1% Tween 20) buffer, and blocked with 3% bovine serum albumin (BSA; Gold Biotechnology, MO, USA). The membrane was incubated with mouse anti-DTMUV-E protein antibody at a dilution of 1:4,000 or incubated with Strep-Tactin®HRP-conjugated (IBA, Göttingen, Germany) conjugate at a dilution of 1:8,000. After washing with PBS-T, if using mouse anti-DTMUV-E protein, the membrane was further incubated with anti-mouse IgG HRP-conjugated secondary antibody (Agilent technologies, Glostrup, Denmark) at a dilution of 1:1,500. Following additional PBS-T washing, the membrane was developed color using the tetramethylbenzidine (TMB) substrate (Seracare, MA, USA).

**Electron microscopy.** The formation of DTMUV-SPs was further confirmed by Transmission Electron Microscopy (TEM). The SPs were prepared for negative staining using Formvar/carbon supported copper grids. A small droplet of the SPs suspension was applied to the grids and left for 3 mins. The grid was rinsed with distilled water and negatively stained with 2% uranyl acetate for 30 sec. The morphology of SPs was observed using transmission electron microscope (HT7700; Hitachi High-Tech, Ibaraki, Japan), and images were recorded at a magnification of 150,000.

## Development of SP-based ELISA for detection of antibody against DTMUV

The DTMUV-SPs were subjected to checkerboard titration to determine the optimal dilution for the assay. Corning® 96 Well EIA/RIA assay microplate (Corning, NY, USA) were coated with Strep-Tactin® (IBA, Göttingen, Germany) at a 1:1,000 dilution and incubated for 16 hrs. at 4°C. The antibodies against DTMUV in the serum samples were detected using an anti-duck IgG HRP-conjugated secondary antibody (Seracare, MA, USA) at a 1:1,500 dilution. OD at 450 nm (OD450) was measured using a microplate reader (Synergy™ H 1, Agilent Technologies, Glostrup, Denmark) to quantify the results. Additionally, a mouse monoclonal antibody against JEV E protein (Abcam, MA, USA) was used to assess the cross-reactivity of the DTMUV-SP-based ELISA.

## DTMUV-Virus-like particles (VLPs) production

The VLPs of DTMUV were produced by co-transfection of plasmids containing the nonstructural protein with the enhanced green fluorescent protein (GFP) reporter gene (pCMV-WNVrep-EGFP), kindly provided by Dr. Shintaro Kobayashi (Laboratory of Public Health, Faculty of Veterinary Medicine, Hokkaido University), and the plasmids containing the structural proteins of DTMUV (strain KPS54A61; pCMV-C-prM-E), designed according to the method described in previous study [23], into HEK-293T cells. The co-expression of the structural protein encapsulating the viral RNA specifically the NS, along with the GFP was monitored using the Indirect Fluorescent Antibody (IFA) technique, with mouse anti-DTMUV-E protein antibodies and a secondary R-Phycoerythrin anti-mouse IgG antibodies (Jackson ImmunoResearch Lab, Penn., USA). The supernatant from the transfected cells was collected and stored at −80°C. Titrations were conducted using the fluorescent focus method. Ten-fold serial dilutions of VLPs were inoculated onto a monolayer of Vero E6 cells in a corning® 96-well clear flat bottom TC-treated microplate (Corning, NY, USA) and incubated for 2 hrs. at 37°C. After incubation, the cells were washed with PBS, followed by the addition of complete medium (2% FBS) and further incubation for 16 hrs. at 37°C. The number of foci were then assessed under a fluorescent microscope (IX73; Olympus, TK, JP), and the viral titer was calculated as fluorescent focus units per milliliter (FFU/ml).

## Focus reduction neutralization test (FRNT)

The VLPs were incubated with the appropriate dilution of each duck serum sample for 1 hr. at 37°C and subsequently inoculated onto the monolayer of Vero E6 cells, and incubated for 2 hrs. at 37°C. Thereafter, the cells were washed with PBS, followed by the addition of complete medium (2% FBS) and further incubated 16 hrs. at 37°C. The number of foci were counted. Only the sample obtained 80% reduction of foci number compared to no serum control was determined as a positive sample for antibody against DTMUV.

## Statistical analysis

All statistical analyses were conducted using R version 4.4.2 (R Foundation for Statistical Computing, Vienna, Austria) with RStudio® version 2024.12.0 [24,25], an integrated development environment (IDE) that facilitates data analysis, visualization, and programming. The ggplot2, ggpubr, readxl, openxlsx, dplyr, and irr packages were used for data manipulation, visualization, and statistical analyses.

For the analysis of OD450 values between the Negative and Positive Control groups, data normality was assessed using the Shapiro-Wilk test, which indicated a non-normal distribution. Consequently, a non-parametric Mann-Whitney U test was performed to compare the groups, with statistical significance defined as $p < 0.05$. The cut-off value was calculated as the mean OD450 of the Negative Control group plus three standard deviations (SD). Results were visualized using Dot Plots generated with the ggplot2 package.

To evaluate cross-reactivity in the SP-based ELISA, Welch's t-test (unpaired t-test assuming unequal variances) was performed at each concentration to compare OD450 values between anti-DTMUV-E and anti-JEV-E protein antibodies, using the base R function t.test. The ggpubr package was employed to generate line plots and to present data as mean ± standard deviation. Statistical significance was defined as a p-value ≤ 0.05, with differences between groups denoted as significant (*) or non-significant (NS).

Sensitivity and specificity, along with their 95% confidence intervals (CI), were calculated using the prop.test function in base R. Agreement between ELISA and FRNT results was assessed using Cohen's kappa statistic via the irr package, under standard assumptions of independence, mutually exclusive outcomes, and consistent classification.

## Results

### Production of DTMUV subviral particles

DTMUV-SPs were generated using a recombinant plasmid designed to express the prM and E proteins of DTMUV. An OSF-tag was strategically inserted between the C-terminus of prM and the N-terminus of E. The cassette of prM-OSF-E was introduced into pCAGGS vector and designated as pCAGGS_DTMUV_prM-OSF-E (Fig 1A).

HEK-293T cells were transfected with the pCAGGS_DTMUV_prM-OSF-E. Both cell lysate and supernatant of the transfected were harvested at 72 hrs. post transfection. The SPs in the supernatant were concentrated by PEG precipitation. The immunoblotting using anti-DTMUV-E antibody and Strep-Tactin®HRP-conjugated demonstrated protein band with molecular weight of 59 kDa (Fig 1B and S1 Fig). Furthermore, the result of TEM demonstrated approximately 35 nm spherical particles (150,000 magnification) in the cultured medium of the transfected cells (Fig 1C). Thus, these results confirmed the formation of SPs.

### Development of SP-based ELISA

To optimize the SP-based ELISA detection system, the positive and negative duck sera (dilution from 500 to 200,000) were examined. A checkerboard titration was conducted to determine the optimal concentration of antigen SPs (ranging from 10 to 70 µg/mL). The results indicated that the optimum of SPs concentration was 70 µg/mL (7 µg/well) and the optimal dilution of duck sera was 1:100,000.

The cut-off value for the developed SP-based ELISA was established using a total of 68 DTMUV RT-PCR negative samples. However, two negative control samples were excluded from the analysis due to abnormally high OD450 values, as explained in the discussion. Consequently, the cut-off value was calculated based on the average OD450 values of the remaining 66 samples. The analysis of data distribution highlighted the need for an adjusted cut-off calculation method to minimize false positive results. The method used was:

$$\text{Cut-off} = \text{Mean} + 3(\text{SD})$$

The average OD450 value of the negative sera was 0.28, with a SD of 0.122 and 3SD of 0.37 (Fig 2). Consequently, the calculated cut-off value was 0.65. This cut-off was used to assess the status of antibodies against DTMUV in all 300 duck sera. A total of 41 samples (14%) tested positive, while 259 samples (86%) were found negative.

**A**

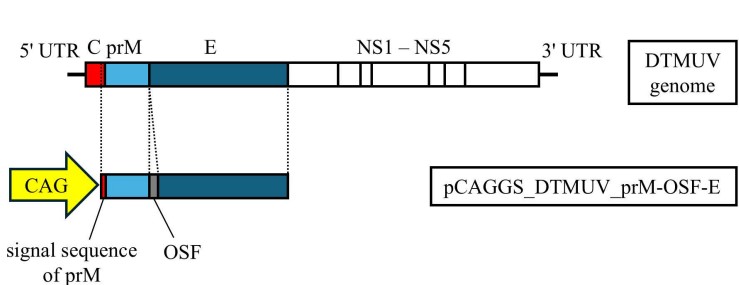

**B**

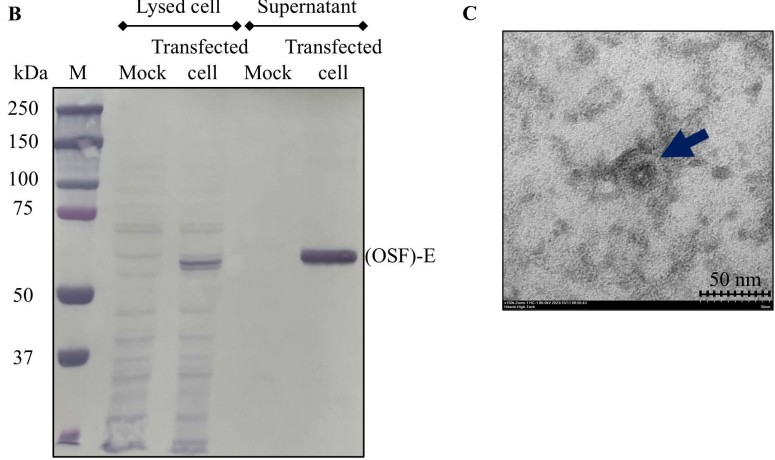

**C**

Fig 1. Demonstration of the production and characterization of SPs of DTMUV. (A) The recombinant plasmid pCAGGS_DTMUV_prM-OSF-E was designed to express DTMUV-SPs. The plasmid contained the CAG promoter, driving the expression of the signal sequence of prM, prM, and E regions, which were linked with OSF tag. (B) HEK-293T cells were transfected with the recombinant plasmid or mock-transfected. Western blotting confirmed the presence of SPs in both the lysed cells and the supernatant. Lane M: protein marker, Lysed cells: Mock-transfected and Transfected cell, and Supernatant: Mock-transfected and Transfected cell. SPs were detected using an anti-DTMUV-E protein antibody (1:4,000 dilution) and an anti-mouse IgG-HRP-conjugated secondary antibody (1:1,500 dilution). A specific band corresponding to the (OSF)-E protein was observed at the expected molecular weight. (C) TEM images of the supernatant from transfected cells reveal SPs. Blue arrows indicate SPs. The scale bar represents 50 nm, with a magnification of 150,000.

To confirm the specificity of the test the SP-based ELISA and the cross-reactivity to other flaviviruses, the assay was evaluated using antibodies against JEV-E and DTMUV-E proteins. The result demonstrated no cross-reactivity of the anti-body against JEV-E protein in the ELISA (Fig 3), confirming the high specificity of the developed test.

## Assessment of specificity and sensitivity of a DTMUV-SP-based ELISA

The confirmation of antibody detection against DTMUV in the serum of infected ducks was conducted through SN using the FRNT technique. To determine the appropriate serum dilution, both positive and negative duck sera were diluted at 1:1,000, 1:10,000, 1:20,000, and 1:40,000, using 100 FFU of VLPs. The results showed that at a dilution of 1:5,000, the number of fluorescent foci was reduced by more than 80% when VLPs were co-incubated with positive sera, compared to the negative control (VLPs without serum). Based on established criteria to ensure assay specificity and minimize cross-reactivity among flaviviruses, a threshold of 80% reduction was applied to define effective neutralization [26,27]. In contrast, negative sera showed no interference, further confirming the specificity of the assay (S2 Fig).

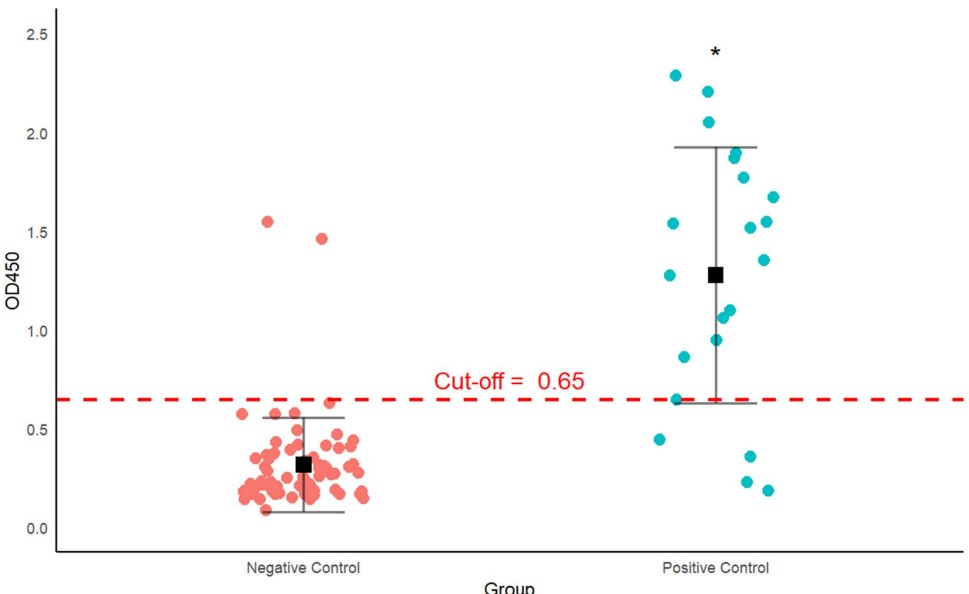

**Fig 2. Dot Plot of OD450 values for Negative and Positive Controls from Duck Sera against Antibodies to DTMUV.** The graph presents the OD450 for two groups: Negative control (n = 68, red) and Positive control (n = 21, blue). The Mean value for each group is represented by the solid black squares, and the error bars (vertical lines) indicate the standard deviation (SD) for each group. The red dashed line represents the cut-off value. Asterisks (*) denote that the difference between the groups was statistically significant (p-value < 0.05), as determined by the Mann-Whitney U test.

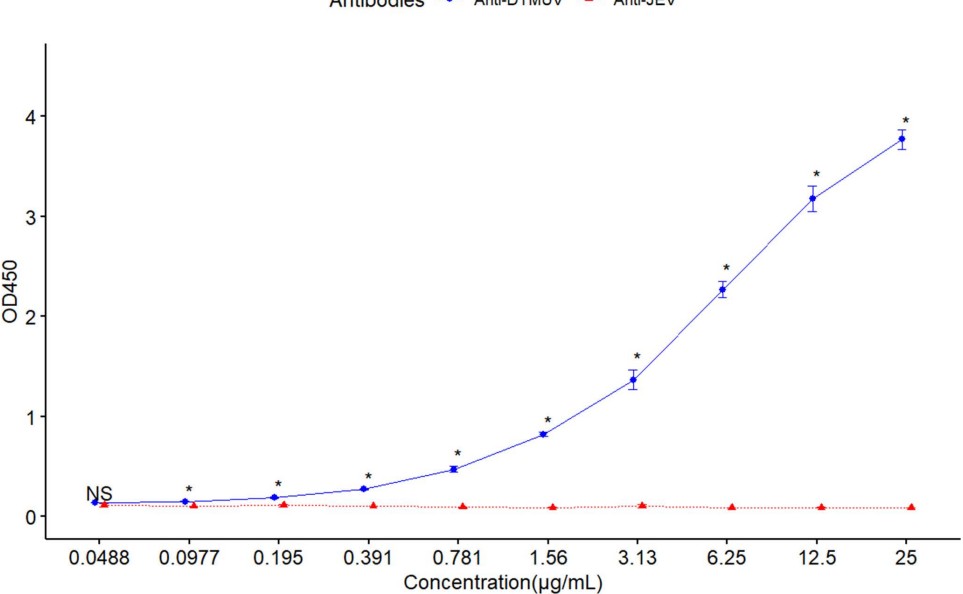

**Fig 3. Analysis of cross-reactivity using SP-based ELISA.** The graph compares OD450 values between two antibodies, Anti-DTMUV and Anti-JEV E proteins, across different concentrations ranging from 0.0488 µg/mL to 25 µg/mL. Welch's t-test (unpaired t-test assuming unequal variances) was used to evaluate statistical differences at each concentration. Points marked with * indicate a statistically significant difference (p-value ≤ 0.05) between the two antibodies at that concentration, while points marked with NS indicate no significant difference (p-value > 0.05). The data points are shown with mean values and standard deviations, with blue representing Anti-DTMUV and red representing Anti-JEV E proteins.

**Table 1. The sensitivity and specificity of the SP-based ELISA.**

| | | FRNT | | |
| | | Positive | Negative | |
|---|---|---|---|---|
| **SP-based ELISA** | Positive | 40 | 0 | 40 |
| | Negative | 0 | 60 | 60 |
| | Total | 40 | 60 | 100 |
| | **Sensitivity** (95% CI) | 100% (89.1-100) | | |
| | **Specificity** (95% CI) | 100% (92.5-100) | | |
| | **Test agreement** | 1.0 | | |

Test agreement was assessed using Kappa statistics

A total of 100 duck serum samples, previously screened using DTMUV-SP-based ELISA, were further assessed for their serological status to DTMUV with the FRNT. These samples were divided into 40 positive and 60 negative sera. The sensitivity and specificity of the SPs based ELISA were calculated to evaluate the diagnostic performance. The result demonstrated that the SP-based ELISA achieved 100% sensitivity and 100% specificity, with a statistical acceptance value of 1.0 (Table 1).

## Discussion

In this study SP-based ELISA was developed employing non-genomic viral particles as antigens. These SPs were derived from the production of recombinant particles designed to mimic the structure of DTMUV, consisting of M and E proteins with the addition of OSF-tag. The E protein plays a key role in stimulating the immune response to DTMUV, while the prM protein is essential for the proper folding of the E protein, ensuring the production of antigens that are highly specific to DTMUV antibodies, thereby reducing cross-reactivity with other flaviviruses [28,29].

During the production of DTMUV-SPs, we performed mammalian cell transfection and confirmed the expression of SPs in the supernatant of the transfected using Western Blot analysis with antibodies specific to the E protein and the Strep-tag. The results demonstrated clear expression of both proteins. Furthermore, structural analysis of the SPs using TEM revealed that the spherical particles with a diameter of approximately 40–45 nanometers. This structure closely resembled whole DTMUV particles [30]. Additionally, the inclusion of the OSF-tag did not affect the production or assembly of the SPs, demonstrating that the expression of the E protein was successful, and the SPs were efficiently assembled into SPs [19,20]. This ensures that the SPs are ready for use in ELISA, allowing for accurate and efficient antibody detection.

The experiment demonstrated that, based on the cut-off value determined from ELISA testing of 66 negative samples, when applied to evaluate 300 duck serum samples, 41 samples tested positive (14%), and 259 samples tested negative (86%). Additionally, serum samples from ducks with confirmed infections by *Riemerella anatipestifer* and Anatid alpha-herpesvirus 1, which are common infections in duck in Thailand, were included in the study to evaluate cross-reactivity. No false-positive results were observed, further demonstrating the high specificity of our ELISA for DTMUV antibodies. This data indicates the ability of ELISA in detecting antibodies against DTMUV in the tested duck population. The detection of positives at this level reflects the effectiveness of the developed ELISA for use in epidemiological studies.

Comparing to the results of the ELISA with the SN test, it was found that the SP-based ELISA demonstrated high diagnostic accuracy. The SP-based ELISA demonstrated 100% sensitivity and specificity, with a perfect statistical acceptance value of 1.0, confirming its reliability as a diagnostic tool. This demonstrates that the DTMUV-SP-based ELISA can accurately detect the antibodies against DTMUV without yielding false negatives or false positives, when compared to other ELISA [11–15]. Moreover, using SPs with addition of Strep-Tactin®, serving as an intermediary between the plate and the

antigen in the ELISA, further addressed the issue of directly coating antigens on the surface of ELISA plate. Direct coating often leads to changes in the conformation of the antigen upon contact with the surface, reducing the efficiency of antibody binding and thereby lowering the sensitivity and specificity of the test. Applying SPs with Strep-Tactin® in the ELISA maintains the 3D structure of the antigen protein, enhancing the accuracy of antibody detection. Additionally, it reduced structural changes in the antigen caused by direct surface contact [19]. This information suggests that SP-based ELISA is a reliable tool that can be effectively used for diagnosis of DTMUV infection.

Additionally, the DTMUV-SP-based ELISA demonstrated a moderate ability in detecting low concentrations of the antibodies present during the early-stage DTMUV infections. When compared to RT-PCR, a molecular technique commonly used for detecting DTMUV infection, the performance evaluation revealed that the developed ELISA had a sensitivity of 81.0% and a specificity of 97.1% in comparison to RT-PCR (S1 Table). A previous study that utilized RT-qPCR and ELISA to detect DTMUV infection in ducks highlighted how RT-qPCR was able to detect viral RNA in the early stages of infection, whereas antibodies had not yet been produced in sufficient quantities for detection by ELISA. This scenario typically occurred during the initial phase of viral infection, when the viral load was sufficient for molecular detection, but the immune response had not yet produced detectable levels of antibodies [30].

A major challenge in serological diagnosis of DTMUV infection is the cross-reactivity with other flaviviruses, such as JEV, due to the structural similarities between these viruses. This cross-reactivity can lead to the detection of antibodies that also respond to other flaviviruses, thereby impacting the accuracy of distinguishing DTMUV infection. DTMUV is known to cause neurological diseases in ducks, leading to outbreaks in duck farms, resulting in mortality and decreased production. Accurate and precise diagnosis is essential for controlling the spread of the virus. To address this issue, we developed an SP-based ELISA to detect DTMUV, which effectively resolved the cross-reactivity problem. The results demonstrated that the developed technique significantly enhanced the specificity of DTMUV antibody detection without cross-reactivity with JEV.

The DTMUV-SP-based ELISA developed in this study holds significant promise for real-world applications, particularly in field-based diagnostics and its potential integration into national or regional avian disease surveillance programs. These applications are especially important given the limitations of conventional diagnostic methods such as serum neutralization tests [10] and RT-PCR [9], while effective for confirming infections, but require advanced laboratory infrastructure, trained personnel, and time-consuming. These challenges often limit their practicality in rural or resource-limited settings, where rapid and decentralized diagnostic tools are urgently needed. In contrast, Our SP-based ELISA exhibits high specificity and sensitivity, allowing for accurate detection of anti-DTMUV antibodies while minimizing cross-reactivity with other flaviviruses. Furthermore, its operational simplicity, affordability, and scalability make it highly suitable for implementation in decentralized laboratories, supporting timely and efficient surveillance under field conditions.

Furthermore, the DTMUV-SP-based ELISA can be readily employed for routine monitoring of duck flocks, particularly during outbreaks or in high-risk areas where rapid detection is critical for containment efforts. Its high specificity significantly reduces the likelihood of false-positive results, enhancing its reliability not only for initial case identification but also for post-vaccination monitoring and longitudinal seroprevalence studies. By enabling accurate and timely detection of DTMUV-specific antibodies, the SP-based ELISA offers an important tool to strengthen avian health surveillance and disease management strategies at both local and national levels.

In addition to routine flock monitoring, the scalability of the DTMUV-SP-based ELISA makes it highly suitable for large-scale surveillance programs, particularly in regions where DTMUV outbreaks have been recurrent. In Thailand, for example, the persistence and increasing severity of DTMUV infections in duck farming areas have raised significant concerns [1–3]. The SP-based ELISA developed in this study provides an effective tool to facilitate early detection and rapid response, supporting national efforts to contain outbreaks and minimize the economic impact on the poultry industry.

Finally, the findings from this study contribute to the broader goal of improving diagnostic preparedness for flavivirus-related diseases. According to the previous studies on establishment of SP-based ELISA of TBEV [19], WNV [20], along with the present study, highlight the potential of SPs as a robust antigenic component in the development of highly

sensitive and specific ELISA-based diagnostics for flavivirus infections. The SP-based ELISA developed here is not only promising for DTMUV detection but also shows practical applicability to other medically important flaviviruses such as TBEV [19], WNV [20], JEV [31], Yellow Fever virus [32], and Dengue virus [33], which are significant global public health concerns and economic challenges.

## Conclusions

This study focuses on the development of a highly specific ELISA for detecting antibodies against DTMUV using subviral particles. The results demonstrate the success of the SP-based ELISA, which exhibited high sensitivity and specificity, effectively identifying DTMUV infections without cross-reactivity to other flaviviruses, such as JEV. In our study, we achieved 100% sensitivity and specificity compared to the Serum Neutralization test, offering a reliable tool for diagnosing DTMUV outbreaks in ducks. This novel diagnostic tool will be invaluable for controlling the spread of DTMUV in duck populations, as it provides a rapid and efficient method for serological screening, which is critical given the significant impact of DTMUV on the poultry industry.

## Supporting information

**S1 Fig. Western blot to illustrate production of DTMUV-SP in HEK-293T cells.** The HEK-293T cells were transfected with the recombinant plasmid or mock-transfected. Western blotting confirmed the presence of SPs in both the lysed cells and the supernatant. Lane M: Protein marker, Lysed cells: mock-transfected and transfected cell, and Supernatant: mock-transfected and transfected cell. SPs were detected using anti-Strep-Tactin® HRP (1:8,000).
(TIF)

**S2 Fig. Comparison of FRNT results under different test conditions.** (A) No serum control, (B) DTMUV-positive serum, and (C) DTMUV-negative serum. All serum samples were diluted to a ratio of 1:5,000. Fluorescence images were captured using a fluorescence microscope at magnification of 100x.
(TIF)

**S3 Fig. Comparison of FRNT results of DTMUV-positive and negative sera with DTMUV-VLPs and WNV-VLPs.** (A, B) No serum control, (C, D) DTMUV-positive serum (utilizing the same serum sample), and (E, F) DTMUV-negative serum (utilizing the same serum sample). All serum samples were diluted to a ratio of 1:5,000. Neutralization assays were performed using FRNT with either DTMUV-VLP (A, C, E) or WNV-VLP (B, D, F) to assess neutralizing activity and potential cross-reactivity. Fluorescence images were captured using a fluorescence microscope at magnification of 100×. An 80% reduction was established as the cut-off to minimize cross-reactivity among flaviviruses and to ensure specificity in neutralization assessment.
(TIF)

**S1 Table. The sensitivity and specificity of the DTMUV-SP-based ELISA compared with RT-PCR.**
(PDF)

**S2 Table. Cross-reactivity test of DTMUV-SP-based ELISA-positive and negative sera to WNV-VLPs evaluated by FRNT.**
(PDF)

**S1 File. Dataset.**
(XLSX)

**S2 File. Dataset.**
(XLSX)

**S3 File. Dataset.**
(XLSX)

**S4 File. Raw_Images.**
(PDF)

## Acknowledgments

We would like to express our gratitude to Prof. Dr. Porntippa Lekcharoensuk and Ms. Nantawan Petcharat: Department of Microbiology and Immunology, Faculty of Veterinary Medicine, Kasetsart University, Thailand for their invaluable support and assistance in providing laboratory equipment, chemicals, and access to the laboratory facilities.

## Author contributions

**Conceptualization:** Thaweesak Songserm, Shintaro Kobayashi, Wallaya Phongphaew.

**Data curation:** Iyarath Putchong.

**Formal analysis:** Iyarath Putchong.

**Funding acquisition:** Wallaya Phongphaew.

**Investigation:** Iyarath Putchong.

**Methodology:** Iyarath Putchong, Sittinee Kulprasertsri, Wallaya Phongphaew.

**Project administration:** Wallaya Phongphaew.

**Resources:** Sittinee Kulprasertsri, Shintaro Kobayashi, Preeda Lertwatcharasarakul.

**Supervision:** Thaweesak Songserm, Preeda Lertwatcharasarakul, Wallaya Phongphaew.

**Validation:** Iyarath Putchong.

**Visualization:** Iyarath Putchong.

**Writing – original draft:** Iyarath Putchong, Wallaya Phongphaew.

**Writing – review & editing:** Iyarath Putchong, Thaweesak Songserm, Sittinee Kulprasertsri, Shintaro Kobayashi, Preeda Lertwatcharasarakul, Wallaya Phongphaew.

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
