## [Decision Letter · Decision Letter 0]

Dear Dr. Phongphaew,

Thank you for submitting your manuscript to PLOS ONE. After careful consideration, we feel that it has merit but does not fully meet PLOS ONE’s publication criteria as it currently stands. Therefore, we invite you to submit a revised version of the manuscript that addresses the points raised during the review process.

**Due to the great difficulty in finding reviewers, my decision is based on a single reviewer. Authors are requested to follow his/her comments in order to improve the manuscript.**

We look forward to receiving your revised manuscript.

Kind regards,

Gianmarco Ferrara

Academic Editor

PLOS ONE

**Journal Requirements:**

1. When submitting your revision, we need you to address these additional requirements. Please ensure that your manuscript meets PLOS ONE's style requirements, including those for file naming. The PLOS ONE style templates can be found at https://journals.plos.org/plosone/s/file?id=wjVg/PLOSOne_formatting_sample_main_body.pdf and https://journals.plos.org/plosone/s/file?id=ba62/PLOSOne_formatting_sample_title_authors_affiliations.pdf 2. Thank you for stating the following financial disclosure: This work was funded by the Office of the Permanent Secretary, Ministry of Higher Education, Science, Research and Innovation (OPS MHESI) under the Research Grant for New Scholar Development Program, grant number RGNS65-038. Additional this work was partially funded by the Faculty of Veterinary Medicine, Kasetsart University, through educational grants and advanced-level graduate research assistantship funding.   Please state what role the funders took in the study.  If the funders had no role, please state: "The funders had no role in study design, data collection and analysis, decision to publish, or preparation of the manuscript." If this statement is not correct you must amend it as needed. Please include this amended Role of Funder statement in your cover letter; we will change the online submission form on your behalf. 3. PLOS ONE now requires that authors provide the original uncropped and unadjusted images underlying all blot or gel results reported in a submission’s figures or Supporting Information files. This policy and the journal’s other requirements for blot/gel reporting and figure preparation are described in detail at https://journals.plos.org/plosone/s/figures#loc-blot-and-gel-reporting-requirements and https://journals.plos.org/plosone/s/figures#loc-preparing-figures-from-image-files. When you submit your revised manuscript, please ensure that your figures adhere fully to these guidelines and provide the original underlying images for all blot or gel data reported in your submission. See the following link for instructions on providing the original image data: https://journals.plos.org/plosone/s/figures#loc-original-images-for-blots-and-gels.   In your cover letter, please note whether your blot/gel image data are in Supporting Information or posted at a public data repository, provide the repository URL if relevant, and provide specific details as to which raw blot/gel images, if any, are not available. Email us at plosone@plos.org if you have any questions.

Reviewers' comments:

Reviewer's Responses to Questions

**Comments to the Author**

1. Is the manuscript technically sound, and do the data support the conclusions?

Reviewer #1: Yes

2. Has the statistical analysis been performed appropriately and rigorously?

Reviewer #1: Yes

3. Have the authors made all data underlying the findings in their manuscript fully available?

Reviewer #1: Yes

4. Is the manuscript presented in an intelligible fashion and written in standard English?

Reviewer #1: Yes

**Reviewer #1: ** Dear Authors,

Thank you for submitting your manuscript, titled "Development of a Highly Specific Enzyme-Linked Immunosorbent Assay for Detection of Antibodies to Duck Tembusu Virus Using Subviral Particles." The study presents significant contributions to the field of veterinary virology by demonstrating an effective and highly specific subviral particle-based ELISA (SP-ELISA) for the detection of Duck Tembusu Virus (DTMUV) antibodies. The use of subviral particles (SVPs) as antigens minimizes cross-reactivity, which has been a long-standing challenge in flavivirus serology. The methodological approach, including the cloning and expression of SVPs in HEK-293T cells, followed by validation of the developed ELISA with a well-characterized sample set (n = 300), is robust. The assay demonstrates high sensitivity and specificity, making it a promising alternative for routine serological surveillance of DTMUV in duck farms.

Further, the manuscript presents novel findings that contribute to diagnostic virology by:

1. Introducing a recombinant SVP-based antigen as a cost-effective alternative to whole-virus ELISA while maintaining specificity.

2. Demonstrating the high diagnostic accuracy of the SP-ELISA, with 100% sensitivity and 98.7% specificity, outperforming conventional ELISAs.

3. Providing a detailed analysis of cross-reactivity, ensuring the reliability of the developed assay against related flaviviruses.

4. Utilizing statistical methods (e.g., Kappa analysis) to validate the agreement between SP-ELISA and virus neutralization tests.

While the study is well-designed and methodologically sound, I recommend some minor revisions to enhance clarity, structure, and depth of discussion.

1. Language & Readability: Some sections can be restructured for clarity and conciseness (see suggested language revisions with page and line numbers below).

2. Statistical Analysis Clarifications: The assumptions of t-tests and Kappa statistics should be briefly mentioned to reinforce statistical rigor.

3. Discussion Refinement: Expand on the potential real-world applications of the SP-ELISA, particularly in field-based diagnostics and integration into surveillance programs.

4. Supplementary Data Enhancements:

o S2 Table (Cross-reactivity Results): Consider adding a footnote explaining the observed cross-reactivity with WNV in some samples.

Suggestions for Language & Readability Enhancements:

1) Line 29-30 (Abstract):

"Serological diagnosis of DTMUV is challenging due to antibody cross-reactivity with other flaviviruses."

2) Line 30-32 (Abstract):

"The membrane precursor and envelope genes of DTMUV (strain KPS54A61) were cloned into a pCAGGS vector with an OSF-tag and transfected into HEK-293T cells to generate subviral particles."

3) Line 39-40 (Abstract):

"The SP-based ELISA exhibited 100% sensitivity and specificity, achieving a perfect agreement score of 1.0 in comparison with the serum neutralization test."

4) Line 47-48 (Introduction):

"Duck Tembusu Virus (DTMUV), a member of the Flaviviridae family and Orthoflavivirus genus, is a major pathogen affecting waterfowl, particularly ducks."

5) Line 55-57 (Introduction):

"These outbreaks severely impacted Thailand’s duck farming industry [1-3]. Additionally, DTMUV can spread to other avian species, including chickens, pigeons, and sparrows[4]."

6) Line 87-88 (Methods):

"the sample size was determined using a prevalence estimation formula for imperfect diagnostic tests, through the Epi-Tool online calculator."

7) Line 118-119 (Methods):

"The ELISA utilized SPs as antigens to enhance specificity and reduce cross-reactivity with other flaviviruses."

8) Line 231-233 (Results):

"DTMUV-SPs were generated using a recombinant plasmid designed to express the prM and E proteins of DTMUV. An OSF-tag was strategically inserted between the C-terminus of prM and the N-terminus of E."

9) Line 270-271 (Results):

"A total of 41 samples (14%) tested positive, while 259 samples (86%) were found negative."

10) Line 310-311 (Discussion):

"In this study SP-ELISA was developed employing non-genomic viral particles as antigens."

11) Line 336-338 (Discussion):

"The SP-based ELISA demonstrated 100% sensitivity and specificity, with a perfect statistical acceptance value of 0.8, confirming its reliability as a diagnostic tool."

Your manuscript is technically sound, methodologically robust, and provides valuable contributions to DTMUV diagnostics, addressing these concerns will further improve the study's clarity, methodology, and discussion. The suggested revisions are required before the manuscript can be considered for publication.

I hope the suggestions would be taken seriously and a well-structured manuscript would be awaited.

Regard

**Do you want your identity to be public for this peer review?** For information about this choice, including consent withdrawal, please see our Privacy Policy

Reviewer #1: **Yes: ** Muhammad Athar Abbas (DVM, PhD)

---

## [Author Response · Author response to Decision Letter 1]

8 May 2025

Dear Reviewer,

We would like to express our gratitude for the constructive feedback and positive evaluation of our work. We appreciate the recommendation for minor revisions to improve the clarity, structure, and depth of discussion. In response, we have revised the manuscript accordingly and addressed all comments in detail as outlined below. All changes are highlighted in the tracked version of the manuscript.

Response to Reviewer – Comment 1: Language & Readability

Response: We appreciate the reviewer’s detailed language suggestions, and we have revised the manuscript accordingly for points 1–10 as recommended.

Comment 1 Line 29-30 (Abstract):

The original sentence “serological diagnosis of DTMUV has been challenging due to antibody cross-reactivity among flaviviruses” has been changed into “Serological diagnosis of DTMUV is challenging due to antibody cross-reactivity with other flaviviruses.”

Comment 2 Line 30-32 (Abstract):

The original sentence “The cassette encoding the membrane precursor and envelope genes of DTMUV (strain KPS54A61), along with an OSF tag, was cloned into a pCAGGS vector and transfected into HEK-293T cells to produce subviral particles” has been changed into “The membrane precursor and envelope genes of DTMUV (strain KPS54A61) were cloned into a pCAGGS vector with an OSF-tag and transfected into HEK-293T cells to generate subviral particles.”

Comment 3 Line 39-40 (Abstract):

The original sentence “Comparison with the serum neutralization test demonstrated that the SP-based ELISA had 100% sensitivity and specificity, with a statistically acceptable value of 1.0” have revised into “The SP-based ELISA exhibited 100% sensitivity and specificity, achieving a perfect agreement score of 1.0 in comparison with the serum neutralization test.”

Comment 4 Line 47-48 (Introduction):

The original sentence “Duck Tembusu Virus (DTMUV) is a virus classified in the family Flaviviridae, genus Orthoflavivirus. This virus causes diseases in waterfowl, particularly ducks” has been corrected into “Duck Tembusu Virus (DTMUV), a member of the Flaviviridae family and Orthoflavivirus genus, is a major pathogen affecting waterfowl, particularly ducks.”

Comment 5 Line 55-57 (Introduction):

The original sentence “These outbreaks caused considerable damage to Thailand’s duck farming industry [1-3]. Furthermore, DTMUV has been shown to spread among other avian, including chickens, as well as resident birds such as pigeons and sparrows [4]” has been revised into “These outbreaks severely impacted Thailand’s duck farming industry [1-3]. Additionally, DTMUV can spread to other avian species, including chickens, pigeons, and sparrows [4]”

Comment 6 Line 87-88 (Methods):

The original sentence “the 87 sample size was calculated using the formula for estimating true prevalence with an imperfect 88 diagnostic test, implemented via the Epi-Tool online calculator [21]” has been revised into “the sample size was determined using a prevalence estimation formula for imperfect diagnostic tests, through the Epi-Tool online calculator [21].”

Comment 7 Line 118-119 (Methods):

The original sentence “The ELISA was designed using SPs as an antigen, to enhance antibody detection specificity and minimize cross-reactivity with other flaviviruses.” has been revised into “The ELISA utilized SPs as antigens to enhance specificity and reduce cross-reactivity with other flaviviruses.”

Comment 8 Line 231-233 (Results):

The original sentence “To generate DTMUV-SPs, we constructed a recombinant plasmid capable of expressing the prM and E protein of DTMUV. The DNA sequence encoding OSF-tag was inserted between the C terminus and N-terminus of prM and E proteins, respectively” has been revised “DTMUV-SPs were generated using a recombinant plasmid designed to express the prM and E proteins of DTMUV. An OSF-tag was strategically inserted between the C-terminus of prM and the N-terminus of E.”

Comment 9 Line 270-271 (Results):

The original sentence “The test results revealed 41 positive samples (14%) 271 and 259 negative samples (86%)” has been revised into “A total of 41 samples (14%) tested positive, while 259 samples (86%) were found negative.”

Comment 10 Line 310-311 (Discussion):

The original sentence “The present study focuses on development of an ELISA technique utilizing subviral particles, non-genomic viral particles, as antigens.” has been revised into “In this study SP-ELISA was developed employing non-genomic viral particles as antigens”.

Comment 11 Lines 336–338 (Discussion):

We would like to clarify that our original sentence was: “the SP-based ELISA achieved 100% for both, with a statistical acceptance value of 1.0” and now has been changed into “The SP-based ELISA demonstrated 100% sensitivity and specificity, with a perfect statistical acceptance value of 1.0, confirming its reliability as a diagnostic tool.” as reviewer’s suggestion.

We are truly grateful for your thorough review and insightful feedback. We apologize for any confusion caused by the initial phrasing and have now corrected the statement to ensure clarity and accuracy.

Response to Reviewer – Comment 2: Statistical Analysis Clarifications

Response:

We appreciate the reviewer’s suggestion to clarify the statistical assumptions. Accordingly, we have revised the Materials and Methods section to include brief descriptions of the assumptions underlying both the t-test and Cohen’s kappa statistic.

Specifically, we now state that data normality was assessed using the Shapiro-Wilk test. Where data were not normally distributed or showed unequal variances, we applied Welch’s t-test or the non-parametric Mann-Whitney U test, as appropriate.

For Cohen’s kappa, we have added that the analysis assumes independent diagnostic methods and mutually exclusive, consistently classified outcomes.

The following sentences have been incorporated into the manuscript to enhance statistical transparency and rigor: “data normality was assessed using the Shapiro-Wilk test, which indicated a non-normal distribution. Consequently, a non-parametric Mann-Whitney U test was performed to compare the groups, with statistical significance defined as p < 0.05.” (Page 10, lines 221–224), “Welch’s t-test (unpaired t-test assuming unequal variances) was performed at each concentration to compare OD450 values between anti-DTMUV-E and anti-JEV-E protein antibodies, using the base R function t.test.” (Page 10, lines 227–229), and “under standard assumptions of independence, mutually exclusive outcomes, and consistent classification.” (Page 10, lines 235–236). In addition, we updated the figure captions for key results to indicate the statistical tests used. Specifically, we added the statistical methods to the captions of Fig. 2 (Dot plot of OD450 values for Negative and Positive Controls from duck sera against antibodies to DTMUV) and Fig. 3 (Analysis of cross-reactivity using SP-based ELISA) to improve clarity and transparency for readers.

Response to Reviewer – Comment 3: Discussion Refinement: Expand on the potential real-world applications of the SP-ELISA, particularly in field-based diagnostics and integration into surveillance programs.

Response:

As suggested, we have expanded the Discussion to better highlight the real-world applications of the SP-based ELISA.

The revised section now emphasizes the assay’s high specificity, sensitivity, and scalability, making it suitable for field-based diagnostics and integration into national or regional avian disease surveillance programs.

We discuss its advantages over conventional methods, its applicability for routine monitoring, post-vaccination surveillance, and large-scale screening, particularly in regions such as Thailand where DTMUV outbreaks have been recurrent.

Finally, we address the broader potential of the assay platform for serological detection of other medically important flaviviruses.

The following paragraphs have been added into the manuscript

“The DTMUV-SP-based ELISA developed in this study holds significant promise for real-world applications, particularly in field-based diagnostics and its potential integration into national or regional avian disease surveillance programs. These applications are especially important given the limitations of conventional diagnostic methods such as serum neutralization tests [10] and RT-PCR [9], while effective for confirming infections, but require advanced laboratory infrastructure, trained personnel, and time-consuming. These challenges often limit their practicality in rural or resource-limited settings, where rapid and decentralized diagnostic tools are urgently needed. In contrast, Our SP-based ELISA exhibits high specificity and sensitivity, allowing for accurate detection of anti-DTMUV antibodies while minimizing cross-reactivity with other flaviviruses. Furthermore, its operational simplicity, affordability, and scalability make it highly suitable for implementation in decentralized laboratories, supporting timely and efficient surveillance under field conditions.

Furthermore, the DTMUV-SP-based ELISA can be readily employed for routine monitoring of duck flocks, particularly during outbreaks or in high-risk areas where rapid detection is critical for containment efforts. Its high specificity significantly reduces the likelihood of false-positive results, enhancing its reliability not only for initial case identification but also for post-vaccination monitoring and longitudinal seroprevalence studies. By enabling accurate and timely detection of DTMUV-specific antibodies, the SP-based ELISA offers an important tool to strengthen avian health surveillance and disease management strategies at both local and national levels.

In addition to routine flock monitoring, the scalability of the DTMUV-SP-based ELISA makes it highly suitable for large-scale surveillance programs, particularly in regions where DTMUV outbreaks have been recurrent. In Thailand, for example, the persistence and increasing severity of DTMUV infections in duck farming areas have raised significant concerns [1-3]. The SP-ELISA developed in this study provides an effective tool to facilitate early detection and rapid response, supporting national efforts to contain outbreaks and minimize the economic impact on the poultry industry.

Finally, the findings from this study contribute to the broader goal of improving diagnostic preparedness for flavivirus-related diseases. According to the previous studies on establishment of SP based-ELISA of TBEV [19], WNV [20], along with the present study, highlight the potential of SPs as a robust antigenic component in the development of highly sensitive and specific ELISA-based diagnostics for flavivirus infections. The SP-based ELISA developed here is not only promising for DTMUV detection but also shows practical applicability to other medically important flaviviruses such as TBEV [19], WNV [20], JEV [31], Yellow Fever Virus [32], and Dengue virus [33], which are significant global public health concerns and economic challenges.” (Revision: Page 17-18, Lines 373–404)

Response to Reviewer – Comment 4: Supplementary Data Enhancements - S2 Table

Response:

Thank you for your helpful suggestion. To improve clarity and provide further context, we have added an explanation to S2 Table explaining the cross-reactivity testing using WNV-FRNT. The added note clarifies that FRNT was conducted using WNV-VLPs and DTMUV-VLPS as a control on randomly selected DTMUV_SP-based ELISA-positive (+) and negative (–) sera. No cross-neutralization against WNV-VLPs was observed in any sample, confirming the specificity of the DTMUV-FRNT assay. We also defined the criteria used to classify positive and negative sera in this context.

The added explanation of table reads:

● Cross-reactivity testing against WNV was performed using the FRNT with WNV-VLPs, with DTMUV-VLPs as a control, on randomly selected DTMUV-SP-based ELISA–positive and negative sera. No cross-neutralization against WNV-VLPs was observed, confirming the specificity of the DTMUV-SP-based ELISA.

(+) = Serum with a reduction in fluorescent foci exceeding 80% was considered positive;

(–) = Serum with a reduction in fluorescent foci of less than 80% was considered negative.

---

## [Editor Report · Decision Letter 1]

Development of a highly specific enzyme-linked immunosorbent assay for detection of antibodies to Duck Tembusu virus using subviral particles

PONE-D-25-03649R1

Dear Dr. Wallaya Phongphaew,

We’re pleased to inform you that your manuscript has been judged scientifically suitable for publication and will be formally accepted for publication once it meets all outstanding technical requirements.

Kind regards,

Gianmarco Ferrara

Academic Editor

PLOS ONE

Additional Editor Comments (optional):

I've personally reviewed the manuscript in the second round of revision due to the unassignment of the reviewer. The authors have addressed all the previous comments
---

## [Editor Report · Acceptance letter]

PONE-D-25-03649R1

PLOS ONE

Dear Dr. Phongphaew,

I'm pleased to inform you that your manuscript has been deemed suitable for publication in PLOS ONE. Congratulations! Your manuscript is now being handed over to our production team.

Kind regards,

on behalf of

Prof. Gianmarco Ferrara

Academic Editor

PLOS ONE